# Probing the pinning strength of magnetic vortex cores with sub-nanometer resolution

Christian Holl[1], Marvin Knol [1], Marco Pratzer[1], Jonathan Chico [2], Imara Lima Fernandes [2], Samir Lounis [2] & Markus Morgenstern [1✉]

Understanding interactions of magnetic textures with defects is crucial for applications such as racetrack memories or microwave generators. Such interactions appear on the few nanometer scale, where imaging has not yet been achieved with controlled external forces. Here, we establish a method determining such interactions via spin-polarized scanning tunneling microscopy in three-dimensional magnetic fields. We track a magnetic vortex core, pushed by the forces of the in-plane fields, and discover that the core (~ $10^4$ Fe-atoms) gets successively pinned close to single atomic-scale defects. Reproducing the core path along several defects via parameter fit, we deduce the pinning potential as a mexican hat with short-range repulsive and long-range attractive part. The approach to deduce defect induced pinning potentials on the sub-nanometer scale is transferable to other non-collinear spin textures, eventually enabling an atomic scale design of defect configurations for guiding and reliable read-out in race-track type devices.

[1] II. Institute of Physics B and JARA-FIT, RWTH Aachen University, D-52074 Aachen, Germany. [2] Peter Grünberg Institut and Institute for Advanced Simulation, Forschungszentrum Jülich and JARA, 52425 Jülich, Germany. ✉email: mmorgens@physik.rwth-aachen.de

Magnetic textures such as vortex cores or skyrmions are key candidates for non-volatile information processing[1–4]. This exploits the texture movement by current pulses that is typically opposed by pinning[5–7]. A detailed understanding of pinning is hence crucial. Moreover, intentional defect configurations can enable reliable positioning of textures in future devices such as race-track memories[1,3,4], i.e., adequate defect configurations could avoid interactions with edges that might annihilate the texture or could locally pin the texture for readout[8,9]. However, previous experiments probing the interaction between texture and defects were either limited in terms of controlled positioning of the texture[10,11] or in terms of spatial resolution[12–19], such that novel methods are urgently needed. In detail, previous experiments probed domain walls[12,14,18], magnetic vortices[13,15,17,19,20] or skyrmions[10] embedded in stripes[14,18], rings[16], islands[13,15,17,19,20] or thin films[10–12] revealing pinning at large, artificial structures (size: 10–100 nm), such as notches[16], holes[15,20] or locally thinned areas of the film[13,14] as well as at intrinsic irregularities, e.g., due to surface roughness[21] or dislocations[11]. Recently, the pinning of skyrmions at single magnetic impurities has also been probed, but without exerting controlled forces[10]. But, so far, experiments were not able to deduce the pinning potential of point defects with the required sub-nm spatial resolution.

Here, we employ spin polarized scanning tunneling microscopy (STM) to study pinning of the curled magnetization of a magnetic vortex core[22,23]. We apply well defined lateral forces by in-plane magnetic fields $\mathbf{B}_\parallel$[24] and independently tune the size of the vortex core by an out-of-plane field $B_\perp$. The latter enables control on the non-collinearity of the core magnetization and, hence, on the strength of exchange energy density $u_{exch}$ in the core. Surprisingly, we find that a vortex core with diameter 3.8 nm within an island of height 10 nm (~$10^4$ Fe-atoms) jumps between defects only a few nm apart. The exact pinning position is deduced by measuring topography and core magnetization simultaneously, revealing an eccentric core pinning ~2 nm away from the next adsorbate. We reproduce the measured core path along several defects via superposing pinning potentials, each consisting of an attractive part with amplitude 200 meV originating from an absent exchange energy and an even stronger repulsive part of unknown origin.

## Results

### Vortex core squeezing and positioning via 3D field.
First, we describe the vortex core squeezing and positioning via the 3D magnetic field. Atomically flat, elliptical Fe islands with vortex configuration are prepared by molecular beam epitaxy on W(110) ("Methods")[23,25]. STM with antiferromagnetic Cr tips[26] records the topography and the spin polarized differential conductance $dI/dV$ simultaneously ("Methods"). Figure 1a displays an overlay of these signals for a typical island. The dark spot in the center indicates the area of the vortex core. The contrast is caused by the spin-polarized $dI/dV$ contribution proportional to the dot product of tip and sample magnetization vectors[27]. Defects are visible (Fig. 1b–d) via the non-magnetic part of $dI/dV$ probing the local density of states[27].

A 3D vector magnet provides $B_\perp$ and $\mathbf{B}_\parallel = (B_x, B_y)$[28], hence, tunes the vortex core size and its position, respectively[23]. Figure 1b–d show $dI/dV$-images of the core at increasing $B_\perp$ opposing the core magnetization. The magnetization can be represented by the normalized out-of plane contribution $m_z$. The Zeeman energy increases and the core diameter shrinks with full width at half maximum (FWHM) of the $m_z$ distribution of $11.0 \pm 0.1$ nm (0 T), $5.48 \pm 0.05$ nm ($-1.2$ T) and $4.34 \pm 0.04$ nm ($-1.5$ T)[23]. This is reproduced by micromagnetic simulations

(Supplementary Note 2) implying a large modification of $u_{exch}$ at the core center due to increasing spin canting: 18 meV/nm$^3$ (0 T), 95 meV/nm$^3$ ($-1.2$ T), 180 meV/nm$^3$ ($-1.5$ T). The $u_{exch}$ tuning enables varying the vortex-defect-interaction for all defects that modify exchange interaction.

Figure 1e reveals the presence of, at least, two types of defects. The 15 pm deep depressions are presumably oxygen adsorbates as remainders from the sample preparation. The 40 pm high protrusions are Cr atoms originating from tip preparation by voltage pulses. Surprisingly, we find that both defects exhibit a quite similar pinning potential (see below). To study the interaction between the vortex core and the defects, we use $\mathbf{B}_\parallel$ to exert a lateral force on the vortex that shifts the core towards a target position. We monitor the deviation from the target due to defect pinning. For a defect-free magnetic cylinder, the core center position $\mathbf{r} = (x, y)$ with respect to the island center is adequately described by the rigid vortex model[29]. It minimizes the potential $E(\mathbf{r}, \mathbf{B}_\parallel) = \frac{1}{2}k(x^2 + y^2) - k\chi_{free}(B_y x + B_x y)$ with $k$ being a proportionally constant and $\chi_{free}$ describing the lateral displacement per field strength. Solving leads to $\mathbf{r}(\mathbf{B}_\parallel) = (\chi_{free}B_y, \chi_{free}B_x)$[17], i.e., the displacement is proportional to $B_\parallel$. Albeit elliptic islands lead to a directional dependence of $k$ and $\chi_{free}$, the core displacement remains largely proportional to $B_\parallel$ (Supplementary Note 4). With additional defects, the potential changes leading to deviations from the regular displacement along a straight path.

### Determining the field induced displacement rate.
In order to determine the field induced displacement rate, firstly at $B_\perp = 0$ T, the vortex core is recorded at three equidistant $\mathbf{B}_\parallel$ (Fig. 1e). The resulting two displacement vectors exhibit equal lengths $\Delta r = 21.5 \pm 0.2$ nm implying a constant displacement rate $\chi(0$ T$) = 1.74$ nm/mT as corroborated in Fig. 2f. In contrast, Fig. 1f shows irregular vortex core motion for $B_\perp = -1.5$ T and 45 equidistant $\mathbf{B}_\parallel$. The core positions are neither equidistant nor along a straight path, but cluster in the vicinity of defects indicating attractive pinning of the core. Remarkably, a vortex core containing ~$10^4$ Fe atoms (diameter: 3.8 nm, height: 10 nm) appears to be pinned close to a single adsorbate. Bending of the core tube is relatively small as verified by micromagnetic simulations (Supplementary Note 5). Long-range interaction of magnetic textures with single adsorbates has previously been shown for skyrmion lattices, where the magnetization has been influenced on a 10-nm scale by switching single coronene molecules[30]. This provides another example of a dramatic influence of single adsorbates on long-range magnetism albeit for a monolayer texture.

The observed pinning of the vortex core naturally reduces the displacement rate $\chi$. To determine the resulting $\chi_{pinned}(B_\perp)$, a second type of experiment is performed. While the vortex core is displaced by 99 equidistant $\mathbf{B}_\parallel$, $dI/dV$ is measured at fixed tip position (Fig. 2a). We target for the identical defect-free path of length ~35 nm along several defects for different $B_\perp$ (Fig. 2b). For $B_\perp = 0$ T, the resulting $dI/dV(B_\parallel)$ features an identical shape as the core shape probed by $dI/dV(\mathbf{r})$ in real space at constant $\mathbf{B}_\parallel$ (gray solid line), i.e., $dI/dV$ is the same for a tip scanning across a fixed vortex core and for a vortex core scanned below a fixed tip by $\mathbf{B}_\parallel$. This confirms, that the large core at $B_\perp = 0$ T barely interacts with the defects. In contrast, the datasets at $B_\perp = -1.2$ T and $-1.5$ T show sudden jumps not appearing in the real space data (Fig. 2d, e). They split the curve into segments of reduced slope $\chi_{pinned}$ due to core pinning. The transitions between the segments correspond to jumps between different pinning sites.

To compare these data with theory, we firstly establish a link between the measured $dI/dV(B_\parallel)$ and the core displacement ("Methods"). The conversion uses the real space $dI/dV(\mathbf{r})$,

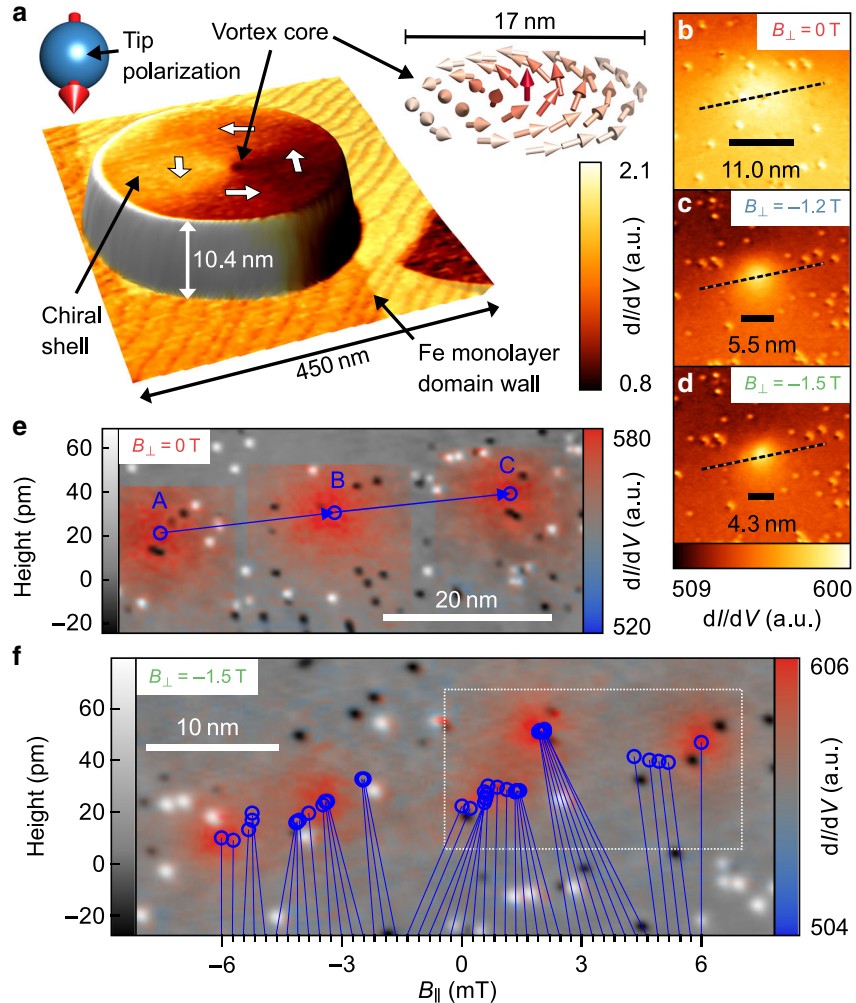

**Fig. 1 Vortex core trajectories. a** Superposition of STM topography (3D representation) and simultaneously acquired spin-polarized d$I$/d$V$ map (color) for an Fe island on W(110), $V = -450$ mV, $I = 0.5$ nA. Insets: sketch of deduced tip magnetization vector (left) and spin configuration of the vortex core (right). **b–d** d$I$/d$V$-images of vortex core at identical contrast, identical scale and different $B_\perp$. The labeled scale bars show FWHM of $m_z$ extracted by core fitting ("Methods"). **e** Superposition of topography (brightness) and three semi-transparent d$I$/d$V$ maps of vortex core (color) after subtracting the signal related to in-plane magnetization (Supplementary Note 3) for $\mathbf{B}_\parallel^A = (20.5, -11.5)$ mT, $\mathbf{B}_\parallel^B = (16, 0)$ mT, and $\mathbf{B}_\parallel^C = (11.5, 11.5)$ mT at $B_\perp = 0$ T. Blue vectors connect the deduced vortex core centers (circles) showcasing the linear core motion. **f** Topography overlaid with vortex core center positions (blue circles) and five selected d$I$/d$V$ maps (color) for 44 equidistant $\mathbf{B}_\parallel$ steps with $\Delta \mathbf{B}_\parallel = (136 - 227)$ μT at $B_\perp = -1.5$ T. The core center positions are connected to the corresponding $B_\parallel$ (lower axis) by lines. The d$I$/d$V$ maps (in-plane magnetization subtracted) correspond to $B_\parallel = -6$ mT, $-3$ mT, 0 mT, 3 mT, 6 mT, $V = -2$ V, $I = 1$ nA. A video of the vortex motion including all 45 d$I$/d$V$ images is available in the supplementary movies. The island size is $255 \times 165 \times 10$ nm³ in **b–f** and $292 \times 210 \times 10.4$ nm³ in **a**.

implicitly assuming an immutable core profile and a straight core path. The core shape indeed exhibits negligible FWHM changes by less than ± 5% along the path (Supplementary Note 6). The motion is not straight (Fig. 2b), but the relatively small excursions imply an error of $\chi_\text{pinned}$ by only 5% (0.3%) at $B_\perp = -1.5$ T ($-1.2$ T) (Supplementary Note 6). Figure 2f–h display the converted data. For $B_\perp = 0$ T, we find one constant slope $\chi = \chi_\text{free}(0$ T$) = 1.8 \pm 0.1$ nm/mT, while, for $B_\perp = -1.2$ T ($-1.5$ T), segments with average slope $\chi_\text{pinned} (-1.2$ T$) = 1.0 \pm 0.1$ nm/mT ($\chi_\text{pinned} (-1.5$ T$) = 0.1 \pm 0.1$ nm/mT) are interrupted by jumps. A small segment with even negative slope appears (Fig. 2h, $B_\parallel = 1$–2 mT) likely originating from a larger sidewards excursion of the core. We deduce a large tuning of the displacement rate ratio $\chi_\text{pinned}/\chi_\text{free} = 100\%$, 42%, and 3% at $B_\perp = 0$ T, $-1.2$ T, and $-1.5$ T, respectively.

To reproduce this, we conduct micromagnetic simulations of an Fe cylinder (diameter: 280 nm, height: 10 nm) with exchange stiffness $A_\text{ex}$, saturation magnetization $M_\text{sat}$, and

uniaxial anisotropies $K_x$, $K_y$ and $K_z$ in the according directions as known from previous experiments[23]. The pinning site is modeled by suppressing $A_\text{ex}$, leading to $u_\text{exch} = 0$, within $1.1 \times 1.1 \times 0.5$ nm³. Note that $A_\text{ex}$ is a parameter independent of the canting of adjacent spins, while $u_\text{exch}$ depends on neighboring spin canting. This defect is moved laterally through the vortex in the island center emulating the vortex movement through the defect by equidistant $\mathbf{B}_\parallel$ ("Methods"). The deduced core path as function of $B_\parallel$ is plotted as solid lines in Fig. 2f–h. It reveals slopes $\chi_\text{pinned}$ around $B_\parallel = 0$ very close to the segment slopes of the experimental data, i.e., we find theoretical $\chi_\text{pinned}/\chi_\text{free} = 96\%$, 40%, and 6% for $B_\perp = 0$ T, $-1.2$ T, and $-1.5$ T, respectively. This strongly suggests that quenching of $A_\text{ex}$ is the origin of pinning.

To corroborate this conjecture, simulations are pursued for defects with changed $K_y$, $K_z$, and $M_\text{sat}$. Figure 3b–d and g–i show resulting pinning potentials deduced from the energy of the vortex core at the corresponding positions ("Methods"). The defect with absent $A_\text{ex}$ features a purely attractive potential with

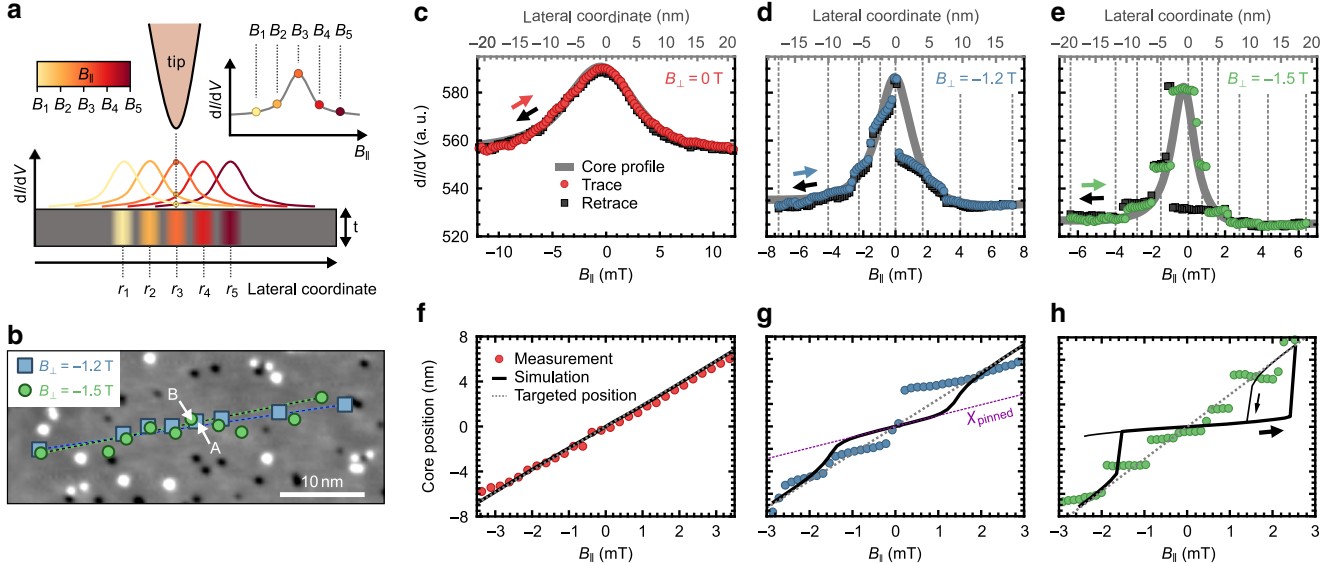

**Fig. 2 Mapping the strength of vortex core pinning. a** Measurement scheme: $B_\parallel$ is stepped equidistantly ($B_1$ to $B_5$) to move the vortex core from $r_1$ to $r_5$, while $dI/dV$ is recorded at fixed tip position. The resulting $dI/dV(B_\parallel)$ displays the core shape in case of a constant core displacement rate. **b** STM topography overlaid with vortex core center positions at two different $B_\perp$ for the $B_\parallel$ highlighted by grey dashed lines in **d**, **e**. The positions are deduced from $dI/dV$ images at the corresponding $B_\perp$. Dotted lines connect start and end point illustrating the target paths. The grayscale colormap represents local height values from −25 pm (black) to 25 pm (white). **c**–**e** $dI/dV$ recorded at the tip position marked by "B" in Fig. 1e for **c**, "A" in **b** for **d** and "B" in **b** for **e** while sweeping $B_\parallel$ at $B_\perp = 0$ T, −1.2 T, and −1.5 T, respectively. The $B_\parallel$ sweep moves the vortex core from "A" to "C" in Fig. 1e, as well as from leftmost to rightmost square or circle in **b**, respectively. The real space $dI/dV$ profiles recorded along the dashed lines in Fig. 1b, 1c, and 1d, respectively, are plotted in gray. The upper and lower axis are linked by the measured average displacement rate, i.e., $(r_C − r_A)/(B_{\parallel,C} − B_{\parallel,A})$ for **c** and, respectively, for **d**, **e**. **f**–**h** Deduced core positions from **c**–**e** assuming a rigid vortex core profile and a straight path (symbols). Solid black lines are micromagnetically simulated core positions for an Fe cylinder (diameter: 280 nm, thickness: 10 nm) with a single pinning site exhibiting $A_{ex} = 0$ for a volume of $1.1 × 1.1 × 0.5$ nm³ at the surface center. Dotted gray lines show simulated displacement without pinning center. The violet line in **g** marks the displacement at rate $\chi_{pinned}$.

an order of magnitude variation in amplitude by $B_\perp$ (Fig. 3g). The defects with changed anisotropy show more complex potentials with amplitudes that are less dependent on $B_\perp$. Tuning $K_y$ toward in-plane anisotropy at the defect (Fig. 3i) leads to repulsion for the out-of-plane oriented area of the vortex core in the absence of $B_\perp$. At finite $B_\perp$, the outer areas of the island are oriented along the field, i.e., out-of-plane, and, thus, only the rim of the vortex core shows in-plane magnetization, such that it gets attracted by such a defect. Using a defect with out-of-plane anisotropy (Fig. 3h) naturally leads to attraction of the core without $B_\perp$, but repels the rim area at finite $B_\perp$, since being the only area without out-of-plane contribution to the magnetization.

For each kind of defect, we simulated the displacement rate $\chi_{pinned}$ around the potential minimum and compared $\chi_{pinned}/\chi_{free}(B_\perp)$ with experimental values (Fig. 3e). The measured trend is quantitatively reproduced for a defect with absent $A_{ex}$, but not for the other types. Using $K_z$ and $K_y$ as unrestricted fit parameters, $\chi_{pinned}/\chi_{free}$ can, at most, be reproduced for one of the three $B_\perp$ within error bars. The optimal fitting, moreover, leads to unrealistically large cumulative anisotropies for a single adsorbate: $V_{defect} × K_z = 86$ meV, $V_{defect} × K_y = 1.1$ eV. These values are orders of magnitude larger than known values of anisotropies, i.e. the bulk anisotropy of Fe would lead to a cumulative anisotropy of 0.2 meV, while the interface anisotropy of Fe to W(110) would lead to 10 meV for the considered defect size[31,32] (see also the ab initio values discussed in Supplementary Note 10). Absent magnetization does barely pin the core at all implying that absent $A_{ex}$ is indeed the main origin of pinning.

**Determining the interaction potential.** Next, we evaluate the precise pinning position of the vortex core center with respect to the closest adsorbate (Fig. 4a). They cluster at 1–2 nm away from

the adsorbate indicating an additional repulsion. Moreover, the offset is mostly directed perpendicular to the target path as expected for an isotropic potential preferentially attracting an object along a line perpendicular to its target path.

To estimate the repelling part of the potential, we employed a fit of 24 subsequent experimental pinning positions (blue dots, Fig. 4b) by adapting three parameters for an identical potential centered at each adsorbate, namely a scaling factor for the axially symmetric ($A_{ex} = 0$)-potential (Fig. 3g, $B_\perp = −1.5$ T) as well as height $A_{rep}$ and FWHM $w_{rep}$ of an axially symmetric Gaussian repelling part (Fig. 4c). The energetic cost of moving the core from the target path toward pinning is firstly calculated without defects via micromagnetic simulations of the vortex energy required to force the core away from its target path. Subsequently, this energy is combined with the pinning potentials yielding the minimum energy position ("Methods"). Figure 4b shows rather good agreement of resulting optimized path (red circles) and measured core positions (blue circles) employing the defect potential of Fig. 4c.

It is a mexican hat with minima located 1.5 nm away from the center as expected from the pinning positions (Fig. 4a). The mexican hat also reproduces the queuing of the core in front of the double defect located above the target path (Fig. 4b). This queuing is markedly different from the slow motion during pinning at a single defect. It cannot be reproduced by overlapping two ($A_{ex} = 0$)-potentials with arbitrary independent positions and, hence, corroborates the mexican hat shape. Naturally, the ($A_{ex} = 0$)-part of Fig. 3g has to be rescaled to compensate for the repelling part, i.e. the ($A_{ex} = 0$)-defect has to be slightly enlarged.

We were not able to pinpoint the origin of the repulsive part. Since it is smaller than the vortex core, it cannot be reproduced by simply changing parameters constantly within a certain area. We

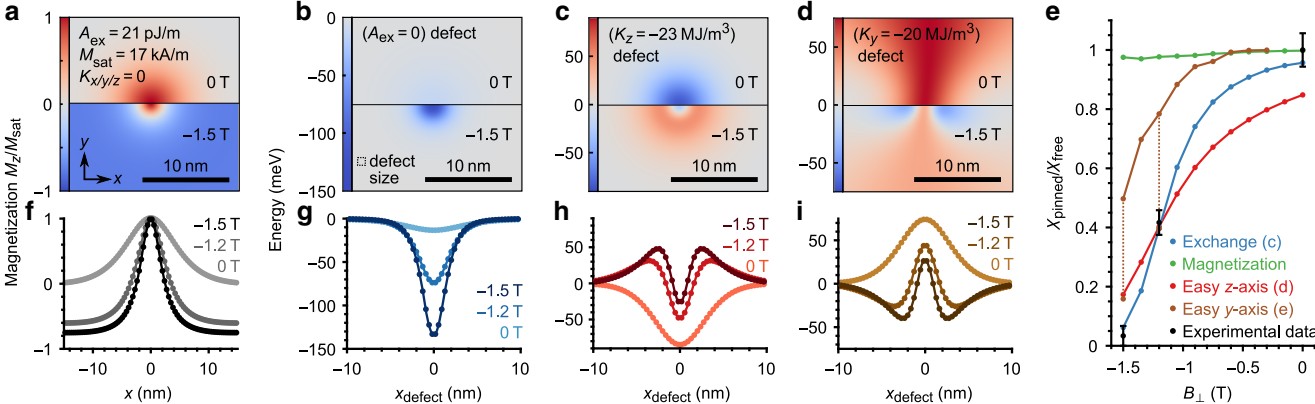

**Fig. 3 Micromagnetic simulation of pinning potentials. a** Scaled, perpendicular magnetization $m_z = M_z/M_{sat}$ of a simulated vortex core in a disk of height 10 nm and diameter 280 nm at $B_\perp = 0$ T (upper half) and $B_\perp = -1.5$ T (lower half) with magnetic parameters indicated. **b–d** Defect potentials for vortex core in meV at different types of magnetic defects, where only the marked parameters are changed with respect to **a** within a central area at the surface of $1.1 \times 1.1 \times 0.5$ nm$^3$ ($3 \times 3 \times 1$ cells). The display type is as in **a**. The spatial dependency of the vortex energy is simulated by scanning the defect through the vortex core ("Methods"). **f–i** Profile lines through the middle of **a–d** (from left to right) covering the 0 T and the $-1.5$ T area separately. An additional profile calculated for $B_\perp = -1.2$ T is added. **e** Simulated displacement rate ratio $\chi_{pinned}/\chi_{free}$ for the vortex core being trapped in the minima of the potentials shown in **g–i**. For the $A_{ex}$ defect, $A_{ex} = 0$ is used and the defect size is adapted to fit the experimental data. For the $K_y$ and $K_z$ defects, we kept the defect size, while $K_y$ and $K_z$ are changed to fit the experimental data as good as possible. For the $K_y$ defect, we show two values for the optimized $K_y = 300$ MJ/m$^3$ and the full line with realistic $K_y = 20$ MJ/m$^3$ connected by dotted lines to the optimized points. For the $M_{sat}$ defect, we use $M_{sat} = 0$ within the same defect volume. The point for $B_\perp = 0$ T of the defect with easy $y$-axis is missing, since the purely repulsive potential did not enable a reliable determination of $\chi_{pinned}$ due to a strong dependence of vortex movement on starting conditions. Experimental data points are deduced from the average slope of the segments such as in Fig. 2f–h with statistical error bars.

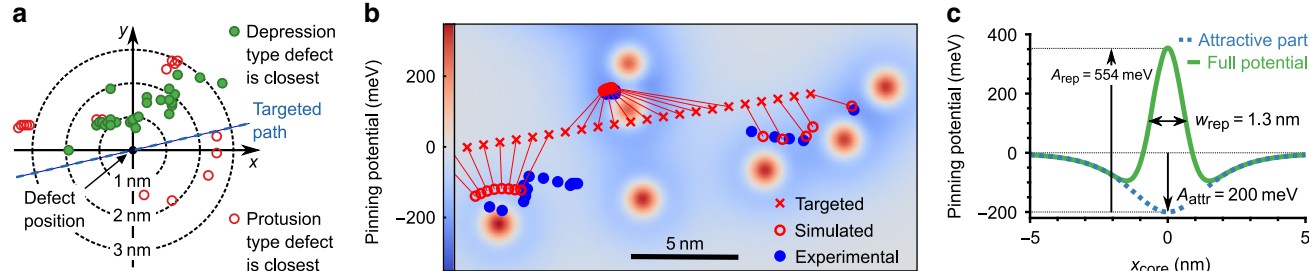

**Fig. 4 Extracting the defect interaction potential. a** Vortex core positions (symbols) with respect to the position of the closest adsorbate, differently colored for the two types of adsorbates. Targeted path (dashed line) reveals that pinning is mostly offset perpendicular to the target path. **b** Simulation of the vortex core path (red circles) within the displayed interaction potential (color map) that superposes the same pinning potential as displayed in **c** centered at each adsorbate. Adsorbate positions are taken from topography (Fig. 1f, white box). Experimental core positions (blue points, Fig. 1f) and target core positions (red crosses) are added. **c** The optimized, axially symmetric single defect potential consisting of an attractive part due to absent exchange energy (Fig. 3g) and a repelling Gaussian part. The three relevant fit parameters are marked.

refrained from optimizing more complex defect structures avoiding the increasing parameter space.

Employing simulations based on density functional theory (DFT), we investigated the impact of single Cr and O adatoms on the magnetic properties of Fe(110). We find remarkably strong changes of the pairwise magnetic exchange interactions $J_{ij}$ ($i$, $j$: atomic sites) affecting up to 70 neighboring Fe atoms (Supplementary Note 10). The summed up change is ~200 meV consisting of similar amounts of weakening and strengthening of $J_{ij}$ due to the oscillatory behavior of the interactions as function of distance. Hence, the sum of changes of $|J_{ij}|$ amounts to 2.5 eV. However, if the vortex core texture is not changed by the defect as implied by the barely changing spin contrast in STM (Supplementary Note 6), the amplitude of the core-adsorbate interaction amounts to only 10–15 meV (Supplementary Note 10). Thus, while the DFT results reveal that single Cr or O adsorbates influence the core path on the 0.5 nm scale (Supplementary Note 10), they do not explain the experiments quantitatively. We speculate that the adsorbate

structure is either different than anticipated or that the adsorbate is accompanied by particular strain fields below the surface accounting for the missing energy. It might be that the adsorbate marks an interface defect via an induced preferential adsorption site.

## Discussion

Our method provides the first quantitative handle on pinning energies of magnetic textures at the sub-nm scale. In principle, it can be applied to different kinds of deliberately placed defects on different types of magnetic islands featuring vortices. It can also be used for other non-collinear textures such as skyrmions or transverse domain walls anticipated to be used in racetrack memories[1,3,4]. Both have been imaged by spin polarized STM[10,27,33]. For skyrmions, additionally the spin canting and, hence, $u_{exch}$ can be tuned by $B_\perp$[34]. Forces on domain walls can be exerted by $\mathbf{B}_\parallel$,[16,35], while skyrmions can be moved by electric currents[5,6], for which respective forces are deduced by combining micromagnetic simulations and an analytic

description via the Thiele equation[36]. Pinning of single skyrmions at defects has already been imaged, e.g. limiting the skyrmion Hall angle[37]. Hence, our method would straightforwardly enable the experimental probing of the so far only theoretically predicted skyrmion-defect interaction strenghts[9,38–40]. Eventually, the method could provide tailoring rules for defect induced guiding of magnetic textures in racetrack memories[8,9].

## Methods

**Preparation**. A W(110) crystal (surface orientation better than 0.1°) is cleaned in ultra high vacuum (UHV) (base pressure: $10^{-10}$ mbar) by repeated cycles of annealing in oxygen atmosphere (partial pressure: $10^{-7}$ mbar) at 1400 °C for 10 min and subsequent flashing to 2200 °C for 10 s. Afterwards, ten pseudomorphic monolayers of Fe are deposited at room temperature by electron beam evaporation from an Fe rod (purity 99.99+%). The sample is then annealed at 710 °C for 20 min leading to the formation of Fe islands such as in Fig. 1a on top of an Fe wetting layer[25]. The method has been optimized to reduce the defect density in the Fe bulk, but we cannot exclude remaining defects, in particular, at the interface of Fe/W(110).

**Spin polarized STM**. The tunneling tip is fabricated from a $0.5 \times 0.5$ mm$^2$ beam of polycrystalline, antiferromagnetic Cr (purity 99.99+%). Tip sharpening employs electrochemical etching by a suspended film of 2.5 M NaOH solution within a PtIr loop that is at potential of 5.5 V with respect to the tip. Etching is stopped at drop off of the lower beam part via differential current detection. The upper part of the beam is immediately rinsed with DI water and glued onto a custom tip holder. The tip is then loaded into the UHV system and, subsequently, into the STM scan head at 6 K[28]. The atomic structure of the tip is optimized during tunneling by voltage pulses (10 V/30 ms) between tip and sample until spin contrast is achieved. Voltage $V$ is applied to the sample. The differential conductance $dI/dV$ is measured by adding a 50 mV RMS sinusoidal voltage (1384 Hz) to the applied DC $V$ and recording the resulting oscillation amplitude of the tunnel current $I$ using a lock-in amplifier. The system enables a 3D magnetic field $\mathbf{B} = (B_x, B_y, B_\perp)$ with out-of-plane component $B_\perp$ up to 7 T and simultaneous in-plane part $\mathbf{B}_\parallel = (B_x, B_y)$ up to 1 T in each in-plane direction[28].

**Adjusting the magnetic field direction**. In order to orient the much stronger out-of-plane field $B_\perp$ avoiding contributions to $B_\parallel$ by misalignment, we used all three magnets to adjust the field direction until the field increase in the resulting direction did not move the vortex core within experimental accuracy. Typically, this required 1.7% of $B_\perp$ to be applied into the nominal $B_\parallel$ direction.

**Micromagnetic simulations**. The program mumax$^3$[41] is used to simulate relaxed magnetization states of an Fe cylinder of height 10 nm and diameter 280 nm with cell size $0.36 \times 0.36 \times 0.5$ nm$^3$. Magnetic parameters are marked in Fig. 3a. Defects are emulated by altered magnetic parameters in $3 \times 3 \times 1$ cells at the top layer. We optimized this size to reproduce the observed pinning restricting ourselves to defects of one layer thickness and a square geometry after crosschecking that the geometry barely changes the pinning behavior as long as the defect volume remains unchanged. For sweeps of $B_\parallel$ with defect, two approximations are employed in order to reduce computational time. Instead of sweeping $B_\parallel = B_{\parallel,\text{target}}$, we keep $B_\parallel = 0$ T and shift the defect through the vortex core by $-\chi_{\text{free}} \cdot B_{\parallel,\text{target}}$ with $-\chi_{\text{free}}$ deduced from a simulation of the vortex with varying $B_\parallel$ but without defects. Second, we crop the simulation area down to $256 \times 256 \times 20$ cells via adding the previously calculated demagnetization field of the neglected area manually. This leads to an effective, spatially varying external magnetic field $\mathbf{B}_{\text{eff}}(\mathbf{r}) = \mathbf{B}_\perp + \mathbf{B}_{\text{demag,exterior}}(\mathbf{r})$. The reasonable validity of these approximations is described in Supplementary Note 7. The resulting core center positions ($m_z$ maxima) as a function of defect position are deduced from spline interpolations of $m_z$ in the layer below the defect. This avoids the more ambiguous evaluation of the partially discontinuous $m_z$ within the surface layer in the presence of defects.

**Vortex core fitting to determine its center position**. To reproduce the experimental spin polarized $dI/dV$ images and, hence, to deduce the vortex core center positions, vortex magnetization patterns are firstly simulated via mumax$^3$. The result is then adapted to the experimental $dI/dV$ image at corresponding $B_\perp$. Therefore, the polar and azimuthal angle of the tip magnetization are optimized using the dot product between sample and tip magnetization vector as $dI/dV$ image contrast. The resulting $dI/dV$ values are additionally offset and scaled to account for the non-spin-polarized $dI/dV$ signal and the unknown amplitude of the spin-polarized $dI/dV$ signal, respectively. Moreover, the vortex core center position is optimized in both lateral directions and the calculated image is slightly scaled laterally to account for inaccuracies of the fit (Supplementary Note 2).

The seven parameters (2 tip magnetization angles, $dI/dV$ offset, $dI/dV$ scaling factor, 2× core position, lateral scaling factor) are fitted towards minimum RMS deviation between the simulated and the measured $dI/dV$ map. The blue circles in Fig. 1e–f as partly also displayed in Fig. 4b and the squares and circles in Fig. 2b are the fitted lateral positions of the vortex core center with each symbol belonging to a fit

of one $dI/dV$ map. The fit error in core center position turns out to be $\leq \pm 0.05$ nm. Fit images, residual images and standard deviations for all fit parameters are given in Supplementary Note 3.

For the superposition of topography and sequences of $dI/dV$ data (Fig. 1e–f), the in-plane magnetization contribution of the fitted $dI/dV$ image is removed from the measured one and, for Fig. 1f, the resulting image is scaled by a Gaussian envelope function for the sake of visibility.

**Conversion from $dI/dV(B_\parallel)$ to core positions**. To calculate a vortex core position from a $dI/dV$ value measured at fixed tip location $\mathbf{r}_0$ but varying $\mathbf{B}_\parallel$, we use the line profile $dI/dV(r')$ of the vortex core measured at constant $\mathbf{B}_\parallel$ (Fig. 1b–d). We first employ the fit procedure as explained in the previous section and then utilize the less noisy profile from the fitted, simulated $dI/dV$ images. Angle of chosen profile line and lateral shift of the profile line with respect to the core center are selected such that the maximum value in $dI/dV(B_\parallel)$ and the $dI/dV$ values at maximum and minimum of $B_\parallel$ (Fig. 2c–e) are reproduced by a straight target path (Supplementary Note 4). The parameter $r'$ is set to zero at maximum $dI/dV$. Using the resulting $dI/dV(r')$, the measured $dI/dV(B_\parallel)$ at $\mathbf{r}_0$ is assigned to a core center position $\mathbf{r}_0 + \mathbf{u} r'(dI/dV(B_\parallel))$ with $\mathbf{u}$ being the unit length vector in the selected profile direction. Principally, there are two possibilities of $r'(dI/dV(B_\parallel))$, left and right from the center of the profile line. They are handled such that the core center always moves to the closer of the two $r'$ and continuously across $\mathbf{r}_0$.

**Calculating vortex core pinning potentials**. To calculate the pinning potentials as displayed in Fig. 3, the parameters are homogeneously changed within $3 \times 3 \times 1$ cells mimicking the defect. Subsequently the defect is moved through the fixed vortex core and the resulting vortex energy is calculated by mumax$^3$. The approximation to move the defect instead of the vortex core is discussed in Supplementary Note 7.

**Simulating the vortex path for multiple defects**. The vortex core position for an immutable core profile is given by minimizing the potential energy $E_{\text{pot}}(\mathbf{r}_{\text{vortex}}) = E_{\text{flex}}(\mathbf{r}_{\text{vortex}} - \mathbf{r}_{\text{target}}) + \sum_{i=1}^{N} E_{i,\text{pin}}(\mathbf{r}_{\text{vortex}} - \mathbf{r}_{i,\text{adsorbate}})$. $E_{\text{flex}}(\mathbf{r}_{\text{vortex}} - \mathbf{r}_{\text{target}})$ is the energetic cost to move the vortex away from its target $\mathbf{r}_{\text{target}}(B_\parallel)$ in the absence of defects. It is deduced from a set of mumax$^3$ simulations fixing the vortex core artificially at different $\mathbf{r}_{\text{vortex}}$. This employs fixing $m_z$ within $4 \times 4$ cells on the surface located away from $\mathbf{r}_{\text{target}}$. The $m_z$ values in that area are set to the values found in the center of the vortex core, if calculated without defects. The vortex core, consequently, moves to a particular $\mathbf{r}_{\text{vortex}}$ with respect to $\mathbf{r}_{\text{target}}$. For this position, we calculate the vortex energy. We checked that the area of fixed $m_z$ leads to negligible changes of the vortex energy (Supplementary Note 8). For sake of simplicity, we approximate the resulting $E_{\text{flex}}(\mathbf{r}_{\text{vortex}} - \mathbf{r}_{\text{target}})$ by an excellently fitting paraboloid (Supplementary Note 8).

The pinning potential of a single adsorbate $E_{i,\text{pin}}(\mathbf{r}_{\text{vortex}})$ is emulated as the sum of a repelling Gaussian and an analytic representation of the attractive part due to a defect with absent $A_{\text{ex}}$. This pinning potential eventually reproduces the profile of Fig. 4c by fitting the core path in Fig. 4b. The analytic representation of the attractive part is derived straightforwardly from an analytic part of the core magnetization profile reading $m_z(r) = (1 - a)/\cosh(|r|/w) + a$ with fit parameters $a$ and $w$[42]. The deduced analytic $u_{\text{exch}}(r)$ is fitted to the result from mumax$^3$ (Fig. 3g) with respect to $a$ and $w$ exhibiting an RMS deviation of only 0.6 meV/nm$^3$ between analytic and micromagnetic representation of $u_{\text{exch}}(r)$ (Supplementary Note 8).

The subsequent fitting of the core path optimizes FWHM and amplitude of the repelling Gaussian as well as a scaling parameter for the attractive, analytic exchange part (Fig. 4c) towards minimizing the RMS of the distances between calculated and measured core center positions (Fig. 4b). Additionally, the start and end point of the target path are varied by up to ±3 nm during the fit with respect to the observed first and last core positions to account for possible pinning at these sites.

## Data availability
The data that supports the plots within this paper and other findings of this study are available from the corresponding author upon request.

## Code availability
All codes used to evaluate the data are available from the corresponding author upon request.

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

## Acknowledgements

We gratefully acknowledge insightful discussions with H.-J. Elmers, S. Blügel, A. Schlenhoff, M. Liebmann and financial support of the German Science Foundation (DFG) via PR 1098/1-1 and of the European Research Council (ERC) under the European Union's Horizon 2020 research and innovation program (ERC-consolidator grant 681405-DYNASORE). We are grateful for the computing time granted by the JARA-HPC Vergabegremium and VSR commission on the supercomputer JURECA at Forschungszentrum Jülich and at the RWTH Aachen supercomputer.

## Author contributions

C.H. provided the idea of the experiment, conducted the measurements under supervision of M.P., performed the micromagnetic simulations, and wrote the first draft of the manuscript, all under supervision of M.M. M.K. contributed to the identification of the defects. J.C. and I.L.F. performed DFT calculations and their analysis under supervision of S.L. M.M. guided the finalizing of the manuscript where all authors contributed.

## Competing Interests

The authors declare no competing interests.
