## [Peer Review File · Nature Communications]

Reviewers' comments:

Reviewer #1 (Remarks to the Author):

The authors present a very interesting work on the pinning of magnetic vortices to atomic surface defects in Fe islands. The work is of relevance for the field of non-collinear spin textures and has implications for application. The manuscript is well written, has a clear message and story line. I like this work very much and have only little criticism.

1) The authors should avoid terms like "novel". I believe it is understood that only novel results, experiments and methods deserve publication in a high impact paper.

2) To my knowledge, the Wiesendanger group published a paper that shows skyrmion rearrangement within a skyrmion lattice upon deposition of single molecules. I guess that this work has in part similar physics, while not explicitly discussed in that paper. To my understanding, this paper should be cited.

3) For me, it was surprising to see this strong pinning by atomic defects, but the authors show convincing data. However, if single impurities manage to pin a vortex core, how about defects inside the film or at the lower Fe/W interface. It is rather difficult to get W clean down to the atomic level. Very often, single C atoms reside on the W surface. Could the authors comment on the cleanliness of the interface? Have they observed pinning at points that show no surface defects? Similarly, the purity of Fe grown by MBE in UHV is limited. Even if you take ideal values of 99.999% purity, the large thickness of 10 nm would result in bulk impurities distributed on a lateral length scale comparable to the experimental images. Can the authors comment on bulk defects?

4) The simulations reproduce qualitatively the pinning but lack a reasonable explanation for the repulsive part. The authors openly state that they have no explanation for this, which I appreciate. Clearly, the exchange energy would not lead to a repulsive part. The anisotropy may, however, result in such a behavior, but it would be present for large and small vortices. Moreover, the needed values of the anisotropy seem to be rather large. Could the same pinning behavior be explained by a combination of pinning vortices at the visible surface defects and pinning at defects inside the layer or the lower interface? This multiple pinning could result in situations, where the energetic minimum of the overall pinning potential from visible and hidden defects is off position of the visible defects, while only the exchange is considered.

Reviewer #2 (Remarks to the Author):

This paper describes an investigation of the microscopic of pinning of vortex cores. Pinning effects in skyrmions, magnetic and superconducting vortices have a large number of implications for basic science and applications. Although there are numerous models for how these objects are pinned, there is relatively little experimental work to understand the microscopic details of the pinning effects. Here the authors use a variety of experimental techniques to study the vortex core trajectories with STM to map out the strength of the pinning. They find evidence that the pinning has a combination of short range repulsion and long range attraction, an effect predicted by certain models. They also provide micromagnetic simulations to confirm the results. Another important aspect of this work is that this method can be used to understand pinning in many other kinds of vortex like textures, creating pinning pinning sites with derived properties, and the ability to control the motion of these objects. Overall this is a very high quality work, well written, and clear, that should have a big impact. I only have a couple of references to suggest. One is a general review of pinning effects in particle like

systems and the other is recent imaging experiments looking at the effects of pinning.

Reichhardt, C. & Reichhardt, C. J. O. Depinning and nonequilibrium dynamic phases of particle assemblies driven over random and ordered substrates: a review. Rep. Prog. Phys. 80, 026501 (2017).

Zeissler, K. et al. Diameter-independent skyrmion Hall angle in the plastic flow regime observed in chiral magnetic multilayers. Nature Commun. 11, 428 (2020).

Reviewer #3 (Remarks to the Author):

The manuscript "Probing the Pinning Strength of Magnetic Vortex Cores with sub-nm Resolution" by Holl and co-authors describes a combined spin polarized STM and micromagnetic calculations study. Overall, the manuscript is of very high quality and describes an approach which may lead to a much better insight into the properties which determine the displacement rate and pinning of magnetic domains in presence of defects. Surprisingly, it is found that the two defects investigated, i.e., single O and Cr atoms on Fe(110), essentially lead to the identical behavior. Given the very different properties of these two species these findings suggest that some hidden and yet unconsidered parameters are responsible for both, the attractive and the repulsive part of the pinning potential. Even though this fact could be stated more clearly at some places of the manuscript, I recommend publication of this nice study after the authors have addressed the points listed below.

1. How do the authors guarantee that the out-of-plane external magnetic field is oriented truly perpendicular to the sample surface (and/or that the in-plane field doesn't contain unintentional out-of-plane contributions)?
2. On page 3, line 8, the authors use the term "vortex depth" but don't adequately define it.
3. Is "quenched exchange" (page 3, 2nd paragraph, line 2 from bottom) a reasonable term?
4. Page 3, results, line 5-7: It is claimed that the vortex core appears as a dark spot. However, Fig. 1 shows that the tip magnetization is not fully out-of-plane but also contains in-plane contributions. Therefore, I would expect that the darkest region in dI/dV does not correspond to the core but is slightly offset.
5. Fig.1: Panels b,c and d shall show equally scaled images.
6. Even though the experimental data were obtained on a sample which contains two types of defects, this doesn't seem to play any role when the data of figures 1 and 2 are analyzed. Why? Later it is revealed that there appears (surprisingly) to be little difference between these two defects. This fact shall be mentioned earlier to avoid confusion.
7. page 5, 2nd par: The variable χ is not properly introduced. (This is only done in the third paragraph).
8. page 5, 3rd par: The sentence "Remarkably, a vortex core containing $\sim 10^4$ Fe atoms (diameter: 3.8 nm, depth: 10 nm) is pinned close to a single adsorbate" shall be replaced by "... appears to be pinned by single adsorbates".

9. page 7, 3rd par, line 3: Why is the uniaxial anisotropy written as $K_x/y/z$? This appears to be confusing.

10. page 7, 3rd par, line 3-4: "The pinning site is modeled by suppressing A_{ex} , respectively u_{exch} , within $1.1 \times 1.1 \times 0.5 \text{ nm}^3$." Two questions related to this sentence: a. Why did the authors chose exactly these spatial dimension? I expect the volume of the defect to be about a factor of 2 smaller, roughly equal to the size of the unit cell. This choice shall be motivated. b. Why do the authors assign two variables to the exchange strength, A_{ex} and u_{exch} ?

11. Some text in Fig. 3 is way too small.

12. In the caption of figure 3 the authors write: "B_perp areas providing purely repulsive vortex core potentials are excluded." This appears to be an unjustified restriction of physical parameters. How do the authors justify this choice?

13. page 7, 4th par: The results of Fig. 3 are inadequately explained and discussed. Especially the findings for modified K_z , K_y reported in Fig. 3c,d,h,i of a partially repulsive interactions deserve a more detailed discussion which not only lists the facts but which gives the reader an intuitive understanding of how K_z , K_y result in repulsive forces between the vortex and the defect.

14. page 8, line 3-1 from bottom: "The optimal fitting, moreover, leads to unrealistically large cumulative anisotropies for a single adsorbate: $V_{\text{defect}} \cdot K_z = 86 \text{ meV}$, $V_{\text{defect}} \cdot K_y = 1.1 \text{ eV}$ (supplement S10)." I suggest to explain here (in the main article) that DFT results in anisotropy energies which are about 3-4 orders of magnitude smaller that the required values.

15. The definition of w_{rep} as marked in fig 4c doesn't seem to make sense. Why don't the authors choose the distance from the max. to the min of the green curve?

16. Supplemental material 10, page 19: The authors write "The Fe(110) substrate with a lattice constant of $a_{lat} = 384 \text{ pm} \dots$ " I don't understand how this number comes about. The Fe lattice constant is 286pm.

Answer to Reviewer #1:

We appreciate that the reviewer finds our results “very interesting”, the manuscript “well written” and that he “like[s] this work very much and [has] only little criticism”. We also appreciate the helpful comments that we considered in detail.

In the following, we respond to each point of the reviewer repeating his/her comments in italic:

1) *“...avoid terms like “novel”.”*

We followed the advice with pleasure and removed the term “novel” to characterize the method everywhere in the manuscript.

2) *“.. the Wiesendanger group published a paper that shows skyrmion rearrangement within a skyrmion lattice upon deposition of single molecules. I guess that this work has in part similar physics, while not explicitly discussed in that paper. To my understanding, this paper should be cited. “*

We assume that the reviewer refers to Nat. Nanotechnol. 9, 1018 (2014). This result is now described in our manuscript for comparison (line 96).

3) *“However, if single impurities manage to pin a vortex core, how about defects inside the film or at the lower Fe/W interface. It is rather difficult to get W clean down to the atomic level. Very often, single C atoms reside on the W surface. Could the authors comment on the cleanliness of the interface? Have they observed pinning at points that show no surface defects? Similarly, the purity of Fe grown by MBE in UHV is limited. Even if you take ideal values of 99.999% purity, the large thickness of 10 nm would result in bulk impurities distributed on a lateral length scale comparable to the experimental images. Can the authors comment on bulk defects?”*

Definitely, we cannot exclude the influence of defects in the bulk of the Fe island. However, we did not observe any pinning at positions without nearby adsorbates such that we conclude that the adsorbate sites are governing the pinning (see answer to next point). We did not find any experiments that mapped the defect contribution in such structures in detail on the 0.1 – 0.01 % level as required and we do not have such experiments available. Hence, currently there is no way to exclude the influence of defects, in particular, at the interface completely. Arguments while still the adsorbate sites are governing the vortex core paths are given in the next answer. We added “The method has been optimized to reduce the defect density in the Fe bulk, but we cannot exclude remaining defects, in particular, at the interface of Fe/W(110).” (line 233).

4) *“The simulations reproduce qualitatively the pinning but lack a reasonable explanation for the repulsive part. The authors openly state that they have no explanation for this, which I appreciate. Clearly, the exchange energy would not lead to a repulsive part. The anisotropy may, however, result in such a behavior, but it would be present for large and small vortices. Moreover, the needed values of the anisotropy seem to be rather large. Could the same pinning behavior be explained by a combination of pinning vortices at the visible surface defects and pinning at defects inside the layer or the lower interface? This multiple pinning could result in situations, where the energetic minimum of*

the overall pinning potential from visible and hidden defects is off position of the visible defects, while only the exchange is considered."

In principle, this is an appealing idea that we also considered. The strongest indication against a scenario of pinning by random interface defects is the excellent simulation of the vortex path in Fig. 4b using an isotropic potential around each adsorbate. In particular, the fact that the double defect above the intended path is blocking the vortex motion rather completely is a strong indication that the potential is indeed centered isotropically at the adsorbate. One can speculate, however, that the adsorbate position is correlated with defects at the interface, e.g., being more strongly adsorbed in an area somehow related to the position of an interface defect. We added "It might be that the adsorbate marks an interface defect via an induced preferential adsorption site." (line 200).

We hope that we could address the points of the reviewer satisfactorily such that she/he can now recommend publication of the manuscript in Nature Communications.

Answer to reviewer #2:

We appreciate that the reviewer regards our work as very "high quality work, well written, and clear, that should have a big impact." We added the proposed references appreciating the hint:

1) Reichhardt, C. & Reichhardt, C. J. O. Depinning and nonequilibrium dynamic phases of particle assemblies driven over random and ordered substrates: a review. Rep. Prog. Phys. 80, 026501 (2017) is now part of the introduction (line 27, ref. [7]).

2) Zeissler, K. et al. Diameter-independent skyrmion Hall angle in the plastic flow regime observed in chiral magnetic multilayers. Nature Commun. 11, 428 (2020) is now part of the conclusions where extrapolating of the method to skyrmions is proposed (line 214, ref. [37]).

We hope that the reviewer can now recommend publication of our manuscript in Nature Communications.

Answer to reviewer #3:

We appreciate that the reviewer finds our results “of very high quality, [...] which may lead to a much better insight into [...] pinning of magnetic domains in presence of defects” and that he “recommend[s] publication of this nice study after the authors have addressed the points ...”. We address these points in the following that led to a significant improvement of the manuscript. We repeat the statements of the reviewer in italic.

- 1) *“How do the authors guarantee that the out-of-plane external magnetic field is oriented truly perpendicular to the sample surface?”*

We adjusted the 3D magnetic field until changing of all three magnetic contributions did not move the vortex anymore, but only changed its size. The required nominal in-plane field was 1.7 % of the applied out-of-plane field. We describe this now in methods as a separate section “Adjusting the magnetic field direction”.

- 2) *“On page 3, line 8, the authors use the term “vortex depth” but don't adequately define it. “*

We changed the phrase to “.. vortex core with diameter of 3.8 nm within an island of height 10 nm ... ” (line 47) .

- 3) *“Is “quenched exchange” (page 3, 2nd paragraph, line 2 from bottom) a reasonable term?”*

We changed the term “quenched” by “absent” throughout the whole text.

- 4) *“Page 3, results, line 5-7: It is claimed that the vortex core appears as a dark spot. However, Fig. 1 shows that the tip magnetization is not fully out-of-plane but also contains in-plane contributions. Therefore, I would expect that the darkest region in dI/dV does not correspond to the core but is slightly offset.”*

This is correct. Indeed, we determined the core center position by comparison of simulated and measured dI/dV images taking the spin canting of the tip into account as outlined in supplementary section S3. The resulting offset of center core position with respect to the darkest area in the dI/dV image is 3.5 nm. We apologize for the sloppy description. We changed it to “The dark spot in the center indicates the area of the vortex core. The contrast is caused by” (line 59).

- 5) *“Fig.1: Panels b,c and d shall show equally scaled images.”*

The three images in Fig. 3b, c, d have the identical scale and contrast as now mentioned in the caption (caption Fig. 1, line 4).

- 6) *“Even though the experimental data were obtained on a sample which contains two types of defects, this doesn't seem to play any role when the data of figures 1 and 2 are analyzed. Why? Later it is revealed that there appears (surprisingly) to be little difference between these two defects. This fact shall be mentioned earlier to avoid confusion.”*

We added “Surprisingly, we find that both defects exhibit a quite similar pinning potential (see below).” directly after mentioning the two types of defects (line 77).

- 7) *“page 5, 2nd par: The variable χ is not properly introduced. (This is only done in the third paragraph).”*

We added into this paragraph “with k being a proportionally constant and χ_{free} describing the lateral displacement per field strength.” (line 83).

- 8) *“page 5, 3rd par: The sentence “Remarkably, a vortex core containing $\sim 10^4$ Fe atoms (diameter: 3.8 nm, depth: 10 nm) is pinned close to a single adsorbate” shall be replaced by “... appears to be pinned by single adsorbates””*

We changed accordingly (line 95).

- 9) *“page 7, 3rd par, line 3: Why is the uniaxial anisotropy written as $K_x/y/z$? This appears to be confusing.”*

We changed to “... uniaxial anisotropies K_x , K_y and K_z in the according directions” (line 128).

- 10) *“page 7, 3rd par, line 3-4: “The pinning site is modeled by suppressing A_{ex} , respectively u_{exch} , within $1.1 \times 1.1 \times 0.5 \text{ nm}^3$.” Two questions related to this sentence: a. Why did the authors chose exactly these spatial dimension? I expect the volume of the defect to be about a factor of 2 smaller, roughly equal to the size of the unit cell. This choice shall be motivated. b. Why do the authors assign two variables to the exchange strength, A_{ex} and u_{exch} ?”*

We used the size of the defect as a fit parameter such that absent exchange energy in this area reproduces the observed pinning. The defect depth is arbitrarily set to one layer of the simulation lattice and we opted for a square defect. We crosschecked that changing the geometry without changing the volume of the defect does not change the pinning. We added “We optimized this size to reproduce the observed pinning restricting ourselves to defects of one layer thickness and a square geometry after crosschecking that the geometry barely changes the pinning behavior as long as the defect volume remains unchanged.” (line 261) . We used the two terms since the exchange energy density u_{exch} that changes spatially due to canting of neighboring spins simplifies direct comparison with other energy densities, while the exchange term A_{ex} is a constant as implemented in the micromagnetic simulations being independent of canting of neighboring spins. We mention the difference now on line 130.

- 11) *“Some text in Fig. 3 is way too small.”*

We increased the size of all subscripts in Fig. 3 to 6 pt.

- 12) *"In the caption of figure 3 the authors write: "B_perp areas providing purely repulsive vortex core potentials are excluded." This appears to be an unjustified restriction of physical parameters. How do the authors justify this choice?"*

The restriction applies only for a single point of Fig. 3e, namely the $B_{\perp} = 0$ T simulation for the easy y-axis defect. For this particular case, we were not able to deduce a reliable χ_{pinned} since the motion surrounding the repulsive potential depends sensitively on the starting conditions. We changed the description to "The point for $B_{\perp} = 0$ T of the defect with easy y-axis is missing, since the purely repulsive potential did not enable a reliable determination of χ_{pinned} due to a strong dependence of vortex movement on starting conditions." (caption Fig. 3, 5th line from below).

- 13) *"page 7, 4th par: The results of Fig. 3 are inadequately explained and discussed. Especially the findings for modified K_z, K_y reported in Fig. 3c,d,h,i of a partially repulsive interactions deserve a more detailed discussion which not only lists the facts but which gives the reader an intuitive understanding of how K_z, K_y result in repulsive forces between the vortex and the defect."*

We added a more detailed discussion on the role of anisotropies. "Tuning K_y towards in-plane anisotropy at the defect (Fig. 3i) leads to repulsion for the out-of-plane oriented area of the vortex core in the absence of B_{\perp} . At finite B_{\perp} , the outer areas of the island are oriented along the field, i.e., out-of-plane, and, thus, only the rim of the vortex core shows in-plane magnetization, such that it gets attracted by such a defect. Using a defect with out-of-plane anisotropy (Fig. 3h) naturally leads to attraction of the core without B_{\perp} , but repels the rim area at finite B_{\perp} , since being the only area without out-of plane contribution to the magnetization." (line 143ff).

- 14) *"page 8, line 3-1 from bottom: "The optimal fitting, moreover, leads to unrealistically large cumulative anisotropies for a single adsorbate: $V_{\text{defect}} \cdot K_z = 86$ meV, $V_{\text{defect}} \cdot K_y = 1.1$ eV (supplement S10)." I suggest to explain here (in the main article) that DFT results in anisotropy energies which are about 3-4 orders of magnitude smaller than the required values. "*

We added a corresponding description "These values are orders of magnitude ..." (line 156) quoting two reviews on anisotropy in thin films and multilayers ([31] J. Phys.: Condens. Matter 16, R603 (2004), [32] Rep. Prog. Phys. 59, 1409 (1996)).

- 15) *"The definition of w_{rep} as marked in fig 4c doesn't seem to make sense. Why don't the authors choose the distance from the max. to the min of the green curve?"*

The potential consists of the part deduced from the simulation of the defect with absent exchange energy and an added repulsive Gaussian. We think that the Gaussian is best described by the height A_{rep} and the FWHM w_{rep} . We apologize that we did not properly introduce these parameters in the previous version of the manuscript as changed now (line

169).

16) *“Supplemental material 10, page 19: The authors write “The Fe(110) substrate with a lattice constant of $a_{lat} = 384 \text{ pm} \dots$ ” I don't understand how this number comes about. The Fe lattice constant is 286 pm .”*

This was a mistake that has been corrected (supplement, page 19, line 16). Indeed, the lattice constant is 286 pm also in the calculation. We apologize for the mistake.

We hope that we have addressed all points of the reviewer satisfactorily such that he/she can now support publication in Nature Communications.

REVIEWERS' COMMENTS:

Reviewer #1 (Remarks to the Author):

In the revised manuscript, the authors answer to all points raised by the referees in a convincing way. All of the minor deficiencies were removed and the paper is fit for publication. I honestly believe that this work on the role of atomic defects in vortex pinning is not only of excellent quality but will have a large impact on the field of non-collinear magnetism. Atomic defects cannot be avoided and thus need to be considered from now on in all analysis in this field. This clearly shows the importance and impact of this work. The presentation of the data and its discussion are of high quality and the general reader is guided step by step to the main conclusion. Thus, I fully support publication in Nature Communications as is.

Reviewer #3 (Remarks to the Author):

The revised version of the manuscript represents a good response to the criticism. To my opinion all comments have been adequately considered and corresponding changes have been made. I support publication in the current state.